# TOOL: AUTOMATICALLY EXTRACTING HARDWARE DESCRIPTIONS FROM PDF TECHNICAL DOCUMENTATION

## Abstract

The ever-increasing selection of microcontrollers brings the challenge of porting embedded software to new devices through much manual work, while code generators are used only in special cases. Since, in practice, usable data is limited to machine-readable formats and the substantial amount of technical documentation is difficult to access due to the print-oriented nature of PDF, we identify the need for a processor to access the PDFs and extract data with a high quality to enable more code generation of embedded software.

In this paper, we design and implement a modular processor for extracting detailed data sets from technical documentation using deterministic table processing for thousands of microcontrollers: device identifiers, interrupt tables, package and pinouts, pin functions, and register maps. Our evaluation of STMicro documentation compares the completeness and correctness of these data sets against existing machine-readable sources with a weighted average of 96.5 % across almost 6 million data points while also finding several issues in both sources. We show that our tool yields very accurate data with only limited manual effort and can enable and enhance a significant amount of existing and new code generation use cases in the embedded software domain that are currently limited by a lack of machine-readable data sources.

## 1 Introduction

With an ever-expanding product catalog of embedded hardware comes the challenge of porting the corresponding hardware-dependent software (HdS) stack to thousands of devices [15]. Hardware vendors typically provide a HdS implementation in the C programming language only However, newer compiled languages and dynamic runtimes, such as [32, 35, 52, 72], bring new programming paradigms and features to resource-limited embedded systems, for which a custom HdS stack is required [15, 29].

Porting HdS to other programming languages is mostly a manual process, where software engineers consult technical documentation to inform design and implementation decisions as well as extract device-specific hardware description data from the documents and convert them into code [15]. However, the technical documentation is often only available as PDF, which complicates the extraction of structured data [16, 48] due to its print-oriented content model [38].

Specifically, porting a HdS stack to a new device requires data for the bootloader (processor, memories, vector table, power management, clock graph), hardware abstraction layer (HAL) (pinout, pin functions, peripherals, register map), device drivers (capabilities, pinout, communication type, register map), and board support (microcontroller, external devices, signal connections, power supply) [15]. Additional data is needed for part evaluation, configuration tools, build systems, testing, and simulation [15]. As a result, the porting process requires a lot of manual labor, which slows down new HdS projects [15].

To alleviate these limitations, some projects use code generators for large parts of their HdS stack [2,32,51–53], with data extracted from machine-readable sources, such as standardized formats [63], proprietary databases from tooling [57], or provided by manually curated datasets [19, 26]. However, the scope and fidelity of the available machine-readable data are usually significantly smaller than what is available in the technical documentation [33,53]. Existing work in the area of document information extraction is focused on generic inputs and cannot provide the domain-specific data found in technical documentation with the necessary accuracy [10,16,48,66].

Combining multiple sources to create the most complete dataset possible in an automated, unsupervised process is thus desirable. As illustrated in Figure 1, the database can then be

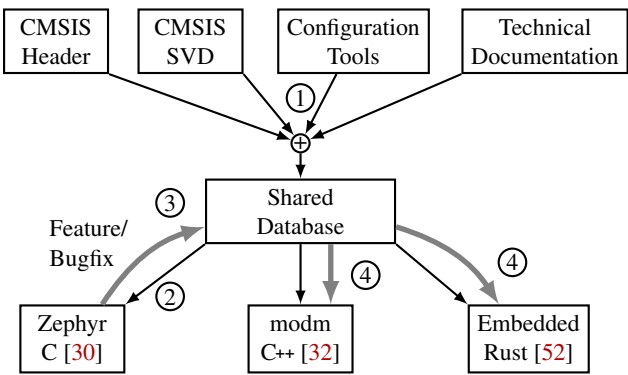

Figure 1: A data processor project ①combining multiple input sources into a shared database without manual supervision promises to reduce overall development efforts. ②A single project using the database can ③add features or fix issues so that ④all other projects benefit from the improvements.

shared among multiple projects, which would significantly reduce development efforts.

Therefore, we identify a research gap in terms of a data processor that connects information extraction of technical documentation with code generation of embedded software. We split this problem into four parts: (i) We first convert the print-oriented PDF technical documentation content into an accessible, structured form. (ii) We then extract and assemble the information relevant to our use cases as determined by domain experts. (iii) We then encode this data into an unambiguous encoding and provide access to it for code generation tasks. (iv) For the evaluation, we compare both the completeness and correctness of the extracted PDF data with the already existing machine-readable counterpart.

**Contributions.** Our tool achieves several contributions. Our processor converts PDF technical documentation to HTML and provides custom parsers for machine-readable data. We apply table processing and text mining paradigms to extract and convert data from the technical documentation in a deterministic process that yields completely reproducible results. We verify the processor functionality by comparing the data extracted from the technical documentation against the existing machine-readable sources. Based on the outcome of this evaluation, internal consistency to establish a method to merge multiple sources and arbitrate conflicts based on qualitative metrics. We also provide a detailed analysis of the quality, trustworthiness, and completeness of each data source that can inform and guide future extraction work. The extracted data is unambiguously encoded as a knowledge graph via a custom ontology to describe the embedded hardware, making it widely accessible to a number of different use cases. Our design is implemented as a highly modular Python project, which will be open-sourced and maintained as part of a broader project.

**Paper Organization.** This paper is organized as follows. In Section 2, we introduce the background relevant to the remainder of this paper. We then describe related work in Section 3 and point out the issues we want to improve on. In Section 4, we describe the design and implementation of our processor, which we then rigorously evaluate against STMicro documentation and data sources, and discuss the results in Section 5. We conclude this paper in Section 6.

## 2 Background

Our tool investigates the quality of hardware description data required for creating embedded software and provides angles to improve the state of the art. Therefore, we now introduce technical documentation formats, table processing, hardware-dependent software, and knowledge modeling.

### 2.1 Technical Documentation

Hardware vendors publish their products' technical documentation as PDF files, which, as a print-oriented format, contain a stream of graphics and text objects placed at precise positions inside the document canvas [38]. As a result, documents render identical on all platforms; however, all semantic and hierarchical information is lost, which makes it difficult to parse and convert automatically [48]. For example, PDFs published by STMicro PDFs contain text-supplementing tables and figures with valuable information. Tables are rendered using vector graphics to draw the cell borders and text characters, as visualized in Figure 2. Extracting structured data from such tables is difficult when relying only on text extraction heuristics [6, 48, 55]. Admittedly, deterministic algorithms consuming vector graphics produce accurate and reliable results [47].

| Pin name (function after reset) | Pin type | I/O structure | Notes | Pin functions | |
|---|---|---|---|---|---|
| | | | | Alternate functions | Additional functions |
| PB13 | I/O | TTa | (4) | SPI2_SCK/I2S2_CK/USART3 _CTS/TIM1_CH1N/ TSC_G6_IO3/EVENTOUT | ADC3_IN5/COMP5_INP/ OPAMP4_VINP/ OPAMP3_VINP |
| PB14 | I/O | TTa | (4) | SPI2_MISO/I2S2ext_SD/ USART3_RTS_DE/ TIM1_CH2N/TIM15_CH1/ TSC_G6_IO4/EVENTOUT | COMP3_INP/ADC4_IN4/ OPAMP2_VINP |

Figure 2: This table excerpt shows the bounding boxes of the individual glyphs in red with their origin marked with a black cross. Inter-document links like the footnote markers in the `Notes` column are marked with a green box [14].

### 2.2 Table Processing

Table processing edits, converts, and formats data from untagged but semi-structured inputs to semantically valuable information [16]. Simple tables render an array of data as a row-column structure of cells [25, 68]; however, complex tables express hierarchical and multi-dimensional information presented in their formatting [16, 25]. The visual rendering of tables includes using different text, separator, and border styles, spanning cells spread over multiple rows and/or columns, cells with multi-line content, and even splitting the entire table into multiple parts to help fit into the presentation medium dimensions, usually a printable page or a digital display [48, 68]. To separate the rendering from its logical structure, the Wang abstract table model [68] defines an indexing relation as a partial function $\delta$ that uniquely maps the multi-dimensional header structure to a table cell resulting in

an *attribute-value* pair [16, 25]. However, any further interpretation and transformation of the tabular data require domain knowledge about the content of the table [48].

## 2.3 Hardware-dependent Software

Hardware-dependent software (HdS) consists of the lowest layers in an embedded system that directly interact with the underlying hardware and provide a portable abstraction to applications on different hardware [5, 29]. In doing so, the HdS can only implement a system functionality *together* with the underlying hardware and would lose its utility without this dependence [15]. Figure 3 illustrates the typical layers of a conceptual and simplified HdS architecture, of which embedded software typically only implements the layers necessary for the scope of the application [5, 29]. Our tool focuses on the hardware description and the HAL, since these layers require the most data and manual effort for porting.

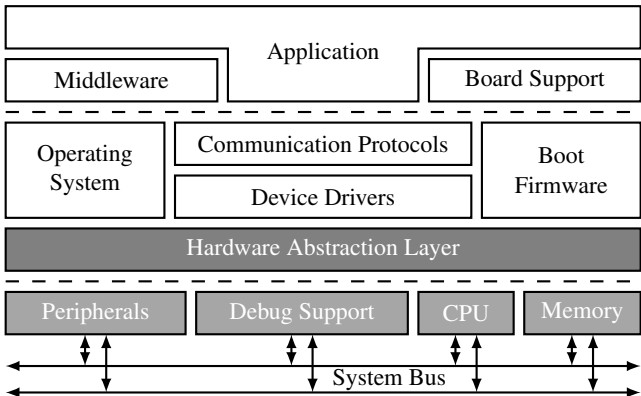

Figure 3: The simplified software stack of a typical HdS architecture with its three layers: software, HdS, and hardware [15]. Our work focuses on the hardware description and hardware abstraction layer (marked gray).

Modern microcontrollers connect a microprocessor to generic internal memories and specialized hardware registers located at specific addresses of an internal bus, known as memory-mapped input/output (MMIO) [15]. To describe the register names, addresses, and types, vendors publish system view description (SVD) files that are the machine-readable equivalent of the register descriptions in the technical documentation [3]. The SVD files are converted into C header files using a closed-source conversion program called `SVDConv` as part of the common microcontroller software interface standard (CMSIS) [65]. This form of MMIO register access is the de-facto standard for all C-based HALs due to the simplicity of the header files that work well with many compilers [3].

Vendors also publish custom tooling for their specific products. For example, the STM32CubeMX [57] tool from STMicro allows a programmer to graphically connect peripheral signals to pins, configure the clock system, estimate power consumption, and enable several middlewares. The tool can then generate a complete C HdS stack for a specific device, which is a pragmatic choice due to its popularity and history as a system programming language [5, 15], but also its main limitation. The tool's internal database contains a machine-readable version of the pin definitions in the technical documentation and is also available separately on GitHub [60].

Apart from C, open-source toolchains also support a number of newer compiled languages, such as C++ [4, 12, 32] and Rust [52, 53]. Optimized runtimes also exist for interpreted languages such as Python [20, 35, 45] and Go [21]. These languages bring new programming paradigms and features to resource-limited embedded systems that are simply not supported by C, especially compile-time code execution and extending the type system. However, all of these new languages must access the underlying hardware with the same MMIO register mechanism. Therefore, they all require the same information to generate their language bindings and support tooling, regardless of what level of abstraction and convenience they provide. However, in practice, all code generation approaches that support new languages are limited to vendor-published, machine-readable, or manually curated data sources.

## 2.4 Knowledge Modeling

Knowledge graphs model facts as edge relations between entities in a graph that can start out very small and, over time, accumulate into a large and rich graph of combined and interlinked facts, which can be used to deduct and validate new knowledge about the domain by providing additional rules on how to interpret entities and relations [22, 27]. The rule set and data graph together constitute a formal representation of domain-specific knowledge, which is called an ontology [22], which can be used to translate the abstract table model via its partial function δ into related facts [66]. We can define the scope and detail of an ontology depending on the extent and quality of the input data and how much additional information we want to query out of it using the same query language and graph algorithms regardless of scale [22].

A concrete implementation of knowledge graphs is the semantic web, which annotates HTML resources with semantics using an XML-based syntax [22]. The data model for semantic web knowledge graphs is the resource description framework, which can be extended with descriptions of semantic rules of increasing computational complexity relative to the reasoning capabilities of a solver [22]. The most used extension is the web ontology language (OWL), which provides basic vocabulary like a datatype hierarchy and pre-defined properties using a description logic with well-understood computational properties that allow reasoning solvers to terminate on all queries [22].

In conclusion, the limitations of the PDF format make accessing its text, figures, and tables difficult, while table

processing allows for working with tabular data in a format-agnostic way if only it were accessible. The technical documentation contains important information for generating HdS in languages other than C that can supplement existing machine-readable sources, whose complex data model we can describe using knowledge graphs. With our tool, we bridge the gaps between these independent areas of research for improved automation and robustness, irrespective of the final use, especially for embedded devices.

## 3 Related Work

After introducing the background as a foundation for the remainder of this paper, we now discuss related work in the areas of information extraction, hardware description pipelines, and embedded software generators.

### 3.1 Document Information Extraction

Extracting structured information from non- or semi-structured inputs is a wide area of research [16]; therefore, we focus on table processing and knowledge modeling. The foundation of table processing is the abstract table model [68], used to decompose a table into its logical structural design, tabular arrangement, and presentation style. Hurst [25] applies this abstract model to tables in documents for the purpose of extracting their information into a semantic model. Embley et al. [16] thoroughly enumerate table input formats, presentation styles, and table processing paradigms. Out of the four table categories in this survey, our work only uses two: Symbolic tables are unambiguously encoded using markup languages such as HTML or XML that separate the table layout from cell content. Their visualization is performed as a separate step, which allows for rendering the same table in different styles [16]. Vector tables are found in PDFs and scalable vector graphics (SVG) and encode the table layout and cell content separately using text instructions for rendering glyphs and graphics instructions for rendering line art [16].

Table processing first requires detecting and locating, then understanding the table's structure and content [16, 28]. For symbolic tables, detection can be as simple as matching on special strings in the content stream [9]. However, for vector tables in PDFs, a common approach is coalescing the bounding boxes of text and graphics into larger clusters [6, 47, 55] and categorizing these into text, figures, and tables. Two common approaches exist: Generic, heuristic algorithms use only the space between text clusters to detect borders [55] if tables provide enough space between the cell content and its borders [6, 47, 49, 50, 55]. A more accurate approach uses the properties of vector graphics to determine table structure directly [47]. However, in practice, both approaches are combined [6, 11, 47] since figures and tables are often composed of a mix of text with implicit whitespace borders and different font properties, alignment, and graphics with

various line types and widths. A particularly robust method is to first locate the table captions via text search and then find the corresponding graphics and text cluster nearby [11].

Information extraction algorithms then align the table structure and content with an externally provided schema to guide the understanding process [16, 48]. Schemas can be user-defined [18] or heuristically obtained [17] and then incrementally merged into a larger ontology [18, 66]. Alternatively, an already existing external ontology [10] or one text-mined from the surrounding text [44, 71] can generate a feasible table schema mapping. The resulting ontology and data are typically modeled via a knowledge graph that can apply various internal reasoning methods in combination with additional external data sources to improve accuracy [22, 27].

To outline how these approaches are applied in practice, we introduce existing hardware data extraction pipelines next.

### 3.2 Hardware Description Data Pipelines

Extracting information from generic documents is a widespread use case for commercial and open-source tools, typically by applying optical character recognition to scanned or photographed documents and heuristic algorithms on the obtained text [28]. However, due to the format ambiguities inherent in such documents, user input is usually required to guide table detection and understanding [28]. For example, Tabula [39] is a popular, open-source Java application that extracts tables into Excel format with a graphical interface to solicit human user input. Khurso et al. [28] compiled even more methods and tools for table extraction in their survey. However, we require tools that extract information specifically from several extensive technical documents related to embedded software and hardware in an automated manner.

Instabuild [40] is a commercial tool to extract device pinout descriptions from screenshots of datasheets but requires human supervision, similar to Tabula. In contrast, uConfig [41] extracts device pinouts automatically using a carefully crafted parser that interprets the text bounding boxes inside the relevant figures but only succeeds for some PDF technical documentation. Finally, Datasheet2SVD [42] uses Tabula to extract the memory map from reference manuals; however, it is limited to work for only two Renesas PDFs documents. However, to the best of our knowledge, none of these projects give any kind of evaluation metric for their accuracy or device coverage or facilitate a simple merging of multiple sources.

Other projects extract information only from machine-readable sources such as CMSIS system view description (CMSIS-SVD), CMSIS-Header, and the STM32CubeMX database [57]. modm-devices [33] accumulates data on device pinouts, pin signal connections, peripheral type and counts, and memory sizes for STM32, SAM, NRF, and AVR microcontrollers, which is used to inform the C++ HAL and toolchain generation in the modm project [32]. The embassy-rs data pipeline [61] does almost the same for generating

| Tool or Project | Open-Source | Maintained | Data Source | Output | Data Scope | Interaction |
|---|---|---|---|---|---|---|
| Tabula [39] | ✓ | ✓ | Any PDF | Excel, CSV | Any table | Supervised |
| Instabuild [40] | ✗ | ✓ | Image of PDF | EDA symbol | Pinout tables | Supervised |
| uConfig [41] | ✓ | ✗ | Datasheet PDF | EDA symbol | Pinout figures | Scripted |
| Datasheet2SVD [42] | ✓ | ✗ | Datasheet PDF | CMSIS-SVD | Register map | Scripted |
| modm-devices [33] | ✓ | ✓ | CMSIS-Header, STM32CubeMX | Custom XML with Python API | Peripherals, pinouts, signals, memories | Scripted with manual patches |
| embassy-rs [61] | ✓ | ✓ | CMSIS-SVD, STM32CubeMX | Custom JSON with Rust API | Peripherals, pinouts, signals, register map | Scripted with manual patches |

Table 1: Comparison of tools and projects that extract hardware description data from PDF and machine-readable sources.

the embassy-rs Rust HAL [53] but is limited to STM32 only. Both tools further store their data in custom formats and do not share any manual data fixes.

In summary, PDF-based tools are limited to extracting very specific data for a limited number of devices, while the most extensive datasets are only generated from the machine-readable sources STM32CubeMX, CMSIS-Header, and CMSIS-SVD. We provide a comparison summary of all these tools and projects in Table 1. To get an overview of the use cases that can consume the hardware description data generated by the pipelines, we present related work in the area of embedded software in the next section.

## 3.3 Generating Hardware-dependent Software

Code generation is an essential tool for HdS design since the limited code space on most devices makes runtime configuration options infeasible [5, 15, 29]. The wide research area of model-driven software engineering includes HAL generation [23, 24], automated testing [69], system modeling [54], and deriving entire software drivers [1] mostly from existing machine-readable sources (cf. Figure 1).

However, in practice, HdS projects only implement a subset of the proposed research. For example, the Linux Zephyr project [30] configures hardware via the DeviceTree [31] interface and then formats it as C pre-processor definitions to be used as an implicit code generator built into the toolchain. The STM32CubeMX configuration tool [57] instead generates its C HAL in an explicit step before compilation, as does modm [32] and Embedded Rust [52] for their respective C++ and Rust HALs. I2CDevLib [26] accumulates manually defined register maps for externally connected devices and provides basic drivers for them. Cyanobyte [19] instead generates device drivers from an abstract dataset so that projects with a custom HAL only need to provide a code template to gain access to all drivers.

Specialized code generators convert CMSIS-SVD register maps [63] found on GitHub [62] into language-specific bindings: SVDConv [65] for generic C, SVD2Rust [51] for Embedded Rust, SVD2Ada [2] for Embedded Ada, and Src-cGen [64] for generic Assembly, C or Clojure definitions.

However, a crowd-sourced effort to significantly improve them is not progressing fast enough [58].

In conclusion, we point out that even though data extraction from tables is a hard but well-understood problem, data pipelines in the embedded software space do not apply these lessons at scale and instead either focus only on extracting only specific data such as pinout detection from documents [40, 41] or only extract data for specific devices [42]. The most extensive pipeline projects [32, 52] eschew documents altogether and only use already machine-readable data [57, 63]. Projects using code generators are therefore limited to the scope of easily accessible data, with a current focus on SVD files, or they are forced to manually build databases to enable their use case [19, 26]. However, several research ideas [1, 54, 69, 70] use very extensive datasets for which a pipeline is missing as of now and, therefore, must derive the required data heuristically or via user input.

## 4 Design and Implementation

The outlined lack of approaches that automatically process and utilize technical documentation convinced us to come up with a design and corresponding tool that transforms and merges multiple data sources into a shared representation annotated with domain-specific semantics. In this section, we give an overview of this design and its specifics. We separate the entire processor into six specialized data pipelines, as visualized in Figure 4. As a result, the individual conversion steps are independent of each other. This modular design also allows for manual or automatic inspection of intermediary data between the stages to assess its quality and tune the conversion process iteratively. Moreover, the data processor can be composed of only those pipelines for which data sources are available, which can then be merged using knowledge graph evolution. For the design overview, we refer to Figure 4. We now introduce the data-processing pipelines in more detail.

**Input Data.** ① Input sources are usually available at vendor websites (e.g., [56, 57]) or GitHub (e.g., [34, 59, 60, 62]).

**PDF → HTML.** The ② PDF to HTML pipeline reverse-engineers the formatting style of the PDF to assign the equivalent HTML semantics to characters, vector graphics, and

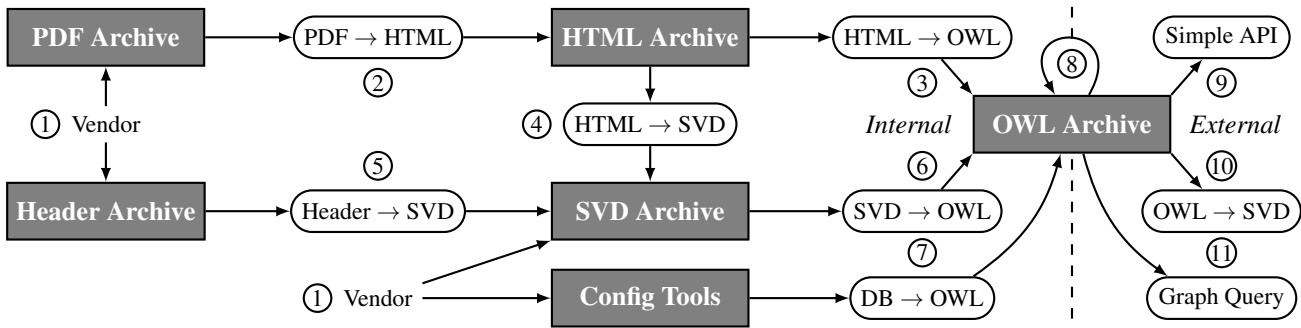

Figure 4: Design overview of our data processor. It relies on several internal pipelines (illustrated on the left side of the dotted line: ②–⑦) and external access methods (shown on the right side: ⑨–⑪). First, ① data sources from the hardware vendor are retrieved. Then, ② the PDF technical documentation is converted to HTML, and ③ the relevant tables contained within are extracted into a knowledge graph (OWL). Additionally, the SVD memory maps are ④ extracted from the HTML, ⑤ CMSIS header files, and vendor-provided SVD files to be ⑥ merged into an optimal representation and stored as a knowledge graph (OWL). Finally, ⑦ the proprietary database (DB) contained in configuration tools is also converted into a knowledge graph (OWL). Then, ⑧ the separate knowledge graphs are evolved into one canonical knowledge graph by a merging strategy that corrects or at least minimizes data conflicts. This final knowledge graph is then ⑨ conveniently accessible via an external Python API, ⑩ converted into specialized formats such as SVD, or ⑪ directly accessible via a knowledge graph query language.

images. To this end, we first abstract the PDF contents into an internal document model via pypdfium2 v1.11 [43], before locating table, figure, and image areas using their caption [11] and vector graphics shape [47], with the remaining areas containing only characters. We then convert each content area separately into an abstract syntax tree (AST) [6,47,55], which describes the logical content hierarchy together with vital formatting metadata, such as text indentation spacing to indicate lexical scope, that further contextualize the individual object semantics. We then unpaginate the content areas by merging these small ASTs into one large AST, which is iteratively modified to better align it to the HTML content model for trivial serialization. The succeeding pipelines can access the documentation now significantly easier in HTML format.

**HTML → X.** The HTML representation provides an abstracted input for the following pipelines ③ and ④. For tables, we provide a simple column-row $(x, y)$ access without regard for table structure, or a partial function $\delta$ that provides attribute-cell pairs as described by the abstract table model [68]. Together with basic text mining via regex matching and substitution, these interfaces are enough to implement all functionality. We now have a library of modular, reusable HTML table queries that operates on the whole document set.

**Header → SVD.** The ⑤ CMSIS header files are converted to CMSIS-SVD files by resolving the C pre-processor macros into numeric values, which are then matched to the peripheral and register names of the C type definitions elements to build our hierarchical memory map. Any memory map definitions in the CMSIS header format can now be parsed and converted.

**X → OWL.** The ⑥ CMSIS-SVD tree format is directly converted into a knowledge graph. The ⑦ STM32CubeMX tool database is encoded as XML and is converted almost

directly without significant effort. The pipelines ③, ⑥, and ⑦ use a manually-defined ontology and serialize to OWL via owlready2 v0.40 [37]. All data is now unambiguously encoded in a common format with shared semantics.

**OWL → OWL.** The knowledge graph is ⑧ evolved by resolving input data conflicts and merging them into a new knowledge graph. This final output of the data processor contains the least flawed data from the input sources.

**OWL → X.** To improve the quality of life for users, we provide several external interfaces (⑨, ⑩, and ⑪) that offer machine-readable access to the extracted knowledge.

These data-processing pipelines manage the automated translation of technical documents and other resources. Having cross-platform use in mind, we entirely implement our design in Python 3.11. Both conceptually and implementationally, our design is very modular. Hence, it can be adapted to other technical documents, vendors, and resources. It further supports replacing or extending specific data pipelines only and promises good utility as pipelines can be (de)selected as needed. The processor will be open-sourced on GitHub under an MPLv2 license, and we will welcome pull requests that augment our existing data pipelines and data sources.

## 5    Evaluation

We now evaluate our implementation effort and establish the quality of the extracted datasets by comparing them against each other. Specifically, we focus on technical documentation by STMicro. In particular, in Section 5.1, we first describe how the pipelines are executed and which input data is converted into intermediary artifacts, before estimating the

pipeline performance in Section 5.2 and the implementation effort of each conversion step in Section 5.3. Subsequently, in Section 5.4, we compare the data we extracted from the technical documentation with their machine-readable counterpart to derive the quality of our automated processing and the accuracy of the provided (technical) documentation. We then discuss our findings in Section 5.5, before concluding with the impact of the evaluation in Section 5.6.

## 5.1 Experimental Setup

The data processor and all auxiliary code are written in Python 3.11 and do not require any special hardware or software setup. We execute all pipelines and measurements on a 2022 Mac-Book Air with an 8-core Apple M2 processor, 16 GB of memory, and 1 TB of storage. For our evaluation, we used the latest STMicro data sources as of March 2023.

**Data Sources.** Before we can start the evaluation, we first need to convert all data sources into a common knowledge graph format. We automatically scraped the technical documentation from the STMicro every day since 16$^{th}$ February 2022, resulting in a total of 1436 PDFs. The ② PDF → HTML pipeline converts the latest revision of 409 PDFs with 156999 pages in total: 70 reference manuals (70 % of the total pages) and 339 datasheets (30 %). The ③ HTML → OWL pipeline then only converts the latest revision of the technical documents, resulting in one knowledge graph for each of the 188 datasheets and 55 reference manuals, ④ as well as 58 SVD files. The ⑤ CMSIS header files for STM32 are converted into 185 SVD files, while ⑥ the 99 CMSIS-SVD files for STM32 are imported as is. The STM32CubeMX database [57] expands into 1316 individual XML files, which the ⑦ DB → OWL pipeline converts into one knowledge graph for each of the 3024 STM32 devices.

**PDF → HTML Quality.** We fine-tuned the accuracy of this pipeline through iterative manual comparison between the PDF and resulting HTML to discover formatting issues and then adapted the code to address them. Nevertheless, we could not fix formatting issues present in the PDF itself with a better algorithm as such actions would have required comprehending the document's content. In these cases, we applied 35 manually created patches with 3289 lines of HTML to almost exclusively add missing table cell borders that caused unrelated cells to be merged. We focus on textual information and tables due to their relevance and intentionally omit figures in the current realization of this pipeline. For our purposes of only extracting structured tabular data, this limitation comes with a good trade-off in the implementation effort and does not impact the data quality and its usefulness.

## 5.2 Processing Times

In the following, we report the average runtime of all pipelines over 10 runs to assess their computational performance. All

conversions are compute-bound and utilize the whole processor with a peak memory usage of about 6 GB. The PDF to HTML conversion takes about 65 min to complete (about 25 ms per page), while the remaining tasks run *much* shorter, only between 2–4 minutes each, for a total of 12 min. These significantly shorter runtimes underline the benefit of converting the PDF to an intermediary HTML format since the algorithmic complexity and the amount of data to process would significantly slow down pipelines depending on the PDF content.

## 5.3 Implementation Effort

Our implementation must be realizable and maintainable with a comparable effort to accessing machine-readable sources directly so that it is a practical alternative. We approximate this metric by the lines of code, as listed in Table 2.

While the technical documentation pipelines consist of only about twice the lines of code as the machine-readable pipelines, they took about 3.5 times as long to implement. The most complex implementation was the PDF → HTML → OWL conversion. We estimate that accessing the technical documentation takes **about three times the effort** overall compared to accessing machine-readable sources. This increased implementation effort is manageable, especially since most of our pipelines can be reused when adding new sources or vendors. Thus, the effort is likely less for future work.

| Pipeline or Task | Lines of Code |
|---|---|
| ① Downloading technical documentation | 105 |
| ② PDF → HTML | + 3105 |
| ③ HTML → OWL | + 1030 |
| ④ HTML → CMSIS-SVD | + 313 |
| **Accessing technical documentation** | **= 4553** |
| ① Downloading machine-readable sources | 124 |
| ⑤ CMSIS Header → CMSIS-SVD | + 481 |
| ⑥ CMSIS-SVD → OWL | + 821 |
| ⑦ STM32CubeMX DB → OWL | + 790 |
| **Accessing machine-readable sources** | **= 2216** |

Table 2: Our data processor requires about twice the lines of code for accessing the technical documentation than the machine-readable sources, as measured by pygount v1.5 [46].

## 5.4 Quality of Extracted Data

We now compare the quality of the data extracted from the technical documentation via pipelines ③–⑦ to existing machine-readable sources, which constitutes the best ground truth available. Given that different data sources use different names to refer to the same entities and relations and aggregate

data into unequally large groups of devices, we only evaluate the completeness of data if we can find an individual device mapping from one source to another (87–93 % of devices). We provide an overview of compared sources and utilized pipelines with their respective sections in Table 3.

| Dataset | Sources | Section |
|---|---|---|
| Device Identifiers | ③ Datasheet vs. ⑦ STM32CubeMX | 5.4.1 |
| Interrupt Vector Table | ③ Reference Manual vs. ⑤ Header | 5.4.2 |
| Package and Pinout | ③ Datasheet vs. ⑦ STM32CubeMX | 5.4.3 |
| Pin Functions | ③ Datasheet vs. ⑦ STM32CubeMX | 5.4.4 |
| Register Descriptions | ④ Reference Manual vs. ⑤ Header vs. ⑥ SVD | 5.4.5 |

Table 3: Compared sources and their respective sections.

### 5.4.1 Device Identifiers

Before we can compare any datasets, we first need to understand which devices it belongs to. This mapping needs to be non-overlapping so that we can have an unambiguous relation from the device identifier to a dataset for comparison. The ⑦ STM32CubeMX database [57] includes a list of 3098 STM32 devices. We removed all devices for which no datasheet or reference manual exits, resulting in 3024 devices. For each ③ datasheet, we produce the list of identifiers via n-fold cartesian product, which generates a total of 14302 STM32 identifiers, over 4 times the STM32CubeMX amount. Using the STM32 identifier schema, we compare the two data sources in Table 4.

The identifier set matches well until the package key is added when the datasheet identifier list explodes with 2806 additional devices. Our implementation does not respect that the pin key, describing the number of pins on a device, interlocks with the package key, and therefore not all combinations can be valid. When we add the temperature key, the missing devices begin to manifest with the full STM32CubeMX identifier list containing 205 devices that cannot be mapped to a datasheet since they are missing these temperature key combinations. Since we could not find any mention of junction temperature in any of the relevant datasheet text or tables, we can only extract 2819 **(93 %) devices** from the documentation. However, these generated identifiers map onto each datasheet and reference manual without any overlaps or gaps.

### 5.4.2 Interrupt Vector Table

The interrupt vector table is extracted from the ③ reference manual and compared with the ⑤ CMSIS header files. We

| Identifier Keys | STM32CubeMX | | Datasheet | |
|---|---|---|---|---|
| Family+Name | 167 | | 171 | +4 |
| +Pin | 656 | | 679 | +23 |
| +Size | 1230 | | 1266 | +36 |
| +Package | 1770 | | 4576 | +2806 |
| +Variant | 1985 | | 6666 | +4681 |
| +Temperature | 2744 | −180 | 9698 | +7134 |
| +Temperature+Variant | 3024 | −205 | 14302 | +11483 |

Table 4: Incrementally appending keys to the STM32 device naming schema shows the datasheet extraction method significantly overestimating the number of produced devices, while some valid temperature keys are missing from datasheets.

can only check for naming conflicts at the same position, but not for completeness since the reference manual contains the maximum population of the vector table, but the header files remove the vectors for peripherals not available on the device. We also ignore datasheets for multi-core devices that implement incomparable, shared interrupt tables, leaving 2751 (91 %) devices to compare.

Our pipeline discovered and assigned the correct table for all devices after normalizing vector names. We were able to match 187 887 out of 190 109 **(98.8 %) compared vector positions**. Of the mismatched positions, 1115 (0.6 %) were missing completely, while 1107 (0.6 %) mostly only differed by a single digit or letter.

### 5.4.3 Package and Pinout

The package and pinout are extracted from a shared table in the ③ datasheet (cf. Figure 2) which contains a package name, the pin positions, and its associated pin name. We compared 2819 devices with a total of 247 756 pins from the ⑦ STM32CubeMX database against the data derived from the datasheets by first finding the correct package, which was successful for 2810 (93 %) devices, and then matching both the name and the position of the pin on the package.

With these fixes, we matched 247 466 **(99.88 %) matched pin positions and names**. We are left with 53 devices that share 290 issues where pins were either missing, added, or unequal in their name and/or position. We investigated each issue manually and classified them into 12 mistakes in the datasheet and 8 issues with the STM32CubeMX database, as detailed in the appendix (Tables 9 and 10). The largest source of errors is the confusion of packages in devices with an optional switched mode power supply feature, which is identified by the variant key and only differs slightly, followed by missing entries or typos in datasheet tables, with plain wrong data being very rare. In no cases did we find bugs in our pipeline implementation or evaluation code, with the packages for 9 devices simply missing from the datasheet.

| Hierarchy Level | Total Memory Map Size | Reference Manual | CMSIS Header | CMSIS-SVD | Overlap | Matching |
|---|---|---|---|---|---|---|
| Peripherals | 55 376 B | 78.8 % | 78.7 % | 87.5 % | 77.2 % | 64.4 % |
| Registers | 1 188 994 B | 79.7 % | 73.8 % | 71.6 % | 74.9 % | 48.3 % |
| Bit Fields | 5 711 619 bit | 73.6 % | 48.2 % | 73.5 % | 60.0 % | 30.6 % |

Table 5: The flat memory map locations are contributed similarly by all three sources, except for the bit fields, where the CMSIS header files are missing a significant amount of data. We call memory locations with more than one source overlapping. If the names of overlapping locations are identical after normalization, we call them matching. The overlap of memory locations gets progressively worse per hierarchy level, with matching locations yielding unacceptably incomplete results.

| Hierarchy Level | Conflict Size | Total Map Size | Conflict-Free Locations | Overlap Map Size | Matching Locations |
|---|---|---|---|---|---|
| Peripherals | 2 748 B | 55 376 B | 95.0 % | 42 752 B | 93.6 % |
| Registers | 35 406 B | 1 188 994 B | 97.3 % | 891 044 B | 96.0 % |
| Bit Fields | 379 180 bit | 5 711 619 bit | 93.3 % | 3 425 903 bit | 88.9 % |

Table 6: The number of conflicts per level and their percentage of conflict-free locations relative to the total size of the memory map or just the locations where two or more sources overlap. The low overlap of just 60 % (cf. Table 5) lowers the bit field numbers even more.

| Hierarchy Level | Resolvable by Majority Vote | Reference Manual + CMSIS Header | CMSIS Header + CMSIS-SVD | CMSIS-SVD + Reference Manual | Matching and Resolved Locations |
|---|---|---|---|---|---|
| Peripherals | 44.1 % | 78.9 % | 19.8 % | 1.3 % | **96.4 %** |
| Registers | 41.7 % | 40.1 % | 15.5 % | 44.4 % | **97.7 %** |
| Bit Fields | 63.0 % | 38.8 % | 22.9 % | 38.3 % | **95.9 %** |

Table 7: Conflicts can be resolved by majority vote only if two sources agree over one other. The reference manual and header files agree the most; however, this pattern becomes less clear at bit-field level. With the voting mechanism, we can increase the accuracy of the memory map, but only for overlapping locations.

### 5.4.4 Pin Functions

In this evaluation step, we compare the pin function tables in the ③ datasheets (cf. Figure 2) with the ⑦ STM32CubeMX database. We must exclude the STM32F1 device family due to a different hardware implementation of pin functions, leaving us with 2692 devices with a total of 1107035 pin functions. However, since the pin function tables in the datasheet contain the union of functions for all devices described, we can only check for conflicts in a signal name against the alternate and additional function index. After the normalization step, we find 1 064 965 **(96.2 %) matching pin-function pairs**, with the remaining 42 070 pairs either missing, added, or wrong in one or the other source. All device families have a relative conflict rate of between 0.8 % and 8.2 %, as listed in the appendix (Table 11). However, the STM32L1 family has a relative conflict rate of 20 %, pointing to a systemic issue in the data, which we investigated manually. A selection of the most prominent patterns is part of the appendix (Table 12).

### 5.4.5 Register Descriptions

To evaluate the register description, we compare three sources: the ④ reference manuals, the ⑤ CMSIS header files, and the ⑥ CMSIS-SVD files. We only compare STM32 devices with an identical MMIO design (excluding ARMv8-M designs), which leaves 2621 (87 %) devices for which all three sources exist. Since these sources have limited device resolution, we perform 183 unique three-way comparisons by checking for name conflicts but not completeness.

Each register description is a tree structure made of peripherals ∋ registers ∋ bit fields. Therefore, we perform the same conflict check at each level. We convert each level into a flat memory map by expanding the peripheral, register, or bit field from the tuple [address, width] into a range of bytes or bits with the corresponding name attached. The total size of each memory map and the amount each source contributes and overlaps are listed in Table 5.

To better understand these numbers, we plot the size of the memory map per three-way comparison, starting with the register level in Figure 5. The figure visualizes how closely our pipeline can reconstruct the register map from the reference manual. The reason for the large discrepancies in the STM32H7 devices is due to the CMSIS header files defining registers related to dual-core management, which are not classified as peripherals in the reference manual or simply omitted in the CMSIS-SVD files. The bit field memory map

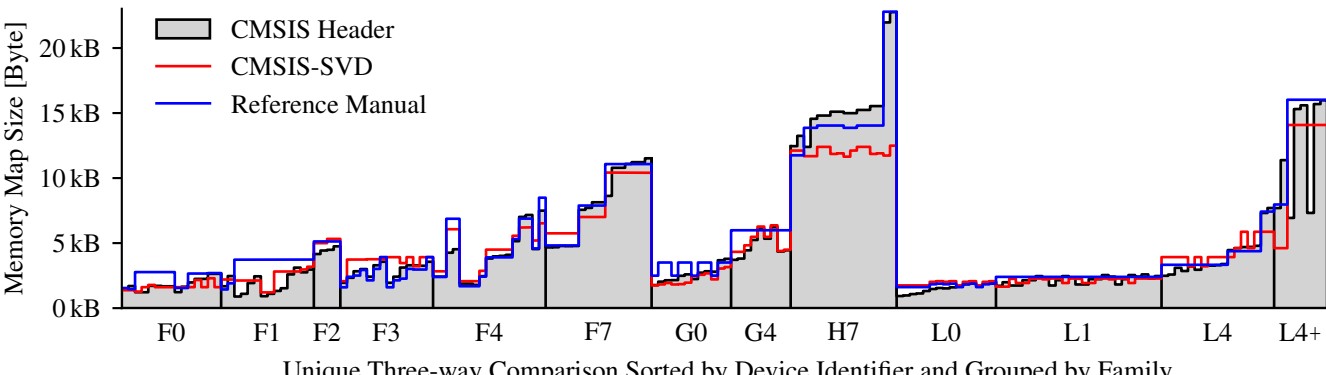

Figure 5: For this graph, we arrange the peripheral memory map sizes for each comparison by device family and alphabetical order. The largest register memory map contains over 23 kB of registers and is placed right next to the smallest with a mere 1810 B. The two peaks in the STM32H7 headers and the two dips in the STM32L4+ headers are caused by the inclusion and omission of the graphics accelerator peripheral GFXMMU, whose register file includes an 8 KiB lookup table. Notice how closely the reference manual matches the CMSIS header for the STM32F3 family, while the SVD files are better suited for STM32G4 devices.

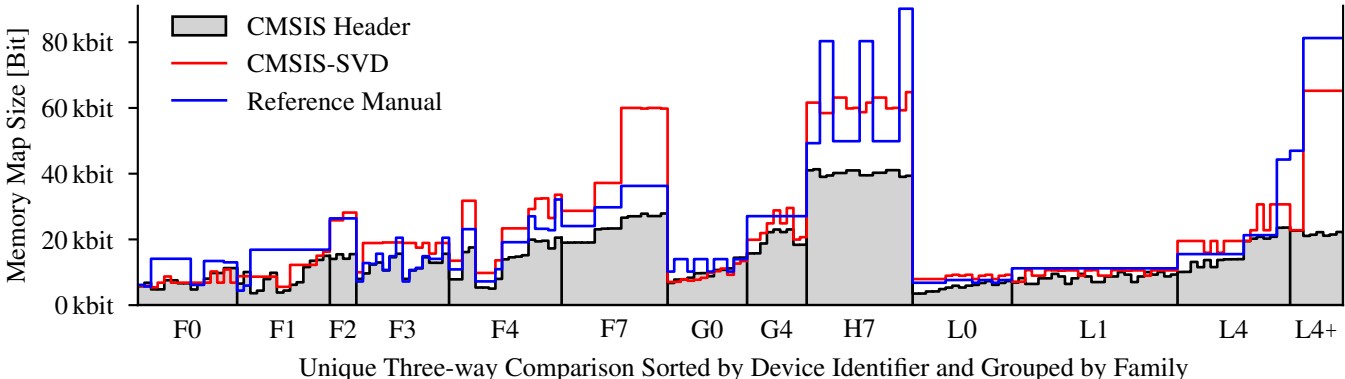

Figure 6: The bit field memory maps extracted from the CMSIS header files are omitting descriptions for registers with only a single large bit field. The effect can be multiplied by arrays, as seen with the STM32L4+ devices. In contrast, the SVD files tend to define all possible bit fields, even if they do not exist on the specific device.

sizes in Figure 6 show a lack of data from the CMSIS header files compared to the reference manuals and SVD files. On inspection, we noticed that the CMSIS header files do not contain bit field definitions for registers that contain only a single integer value since a 32-bit or 16-bit value can be natively constructed using the C `uint32_t` or `uint16_t`, respectively.

We continue our evaluation by highlighting all memory locations whose names do not match after normalization for all three levels resulting in Table 6. Compared to the whole memory map size, the amount of conflict-free locations is fairly high at above 97 % for registers and still 93 % for bit fields. We can improve these results with majority voting, which requires two agreeing sources overruling one other per location. These requirements are fulfilled by about 42–63 % of overlapping memory locations, as shown in Table 7. We can also see significant differences per level in the source combinations that agreed most during the voting process, par-

ticularly on register and bit field level, where the combinations using the reference manuals agree most often, demonstrating the accuracy and usefulness of our pipeline. This simple voting mechanism is sufficient to significantly improve the percentage of conflict-free overlapping memory, resulting in **98 % of registers and 96 % of bit fields**.

When visualizing the relative conflict rate of registers per device family in Figure 7, we discover that both the conflict distribution as well as the majority voting opportunities are not equally distributed among the memory maps. The largest amount of conflicts is attributed to the STM32F7, STM32H7, and STM32L4+ families, which are complex microcontrollers. Simpler devices exhibit fewer conflicts to begin with and a higher share of majority voting. For the bit field conflict distribution shown in Figure 8, these patterns are spread more widely, and the simpler devices have even more opportunities to use majority voting.

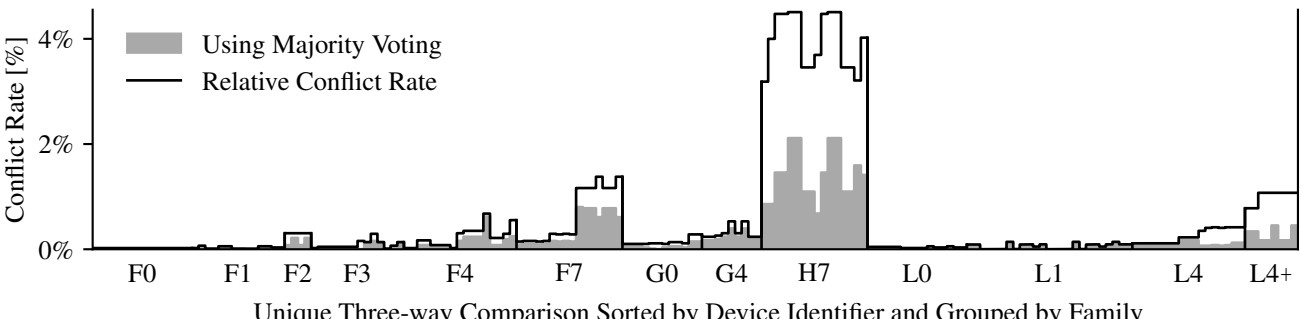

Figure 7: The distribution of register conflicts is not equally distributed, with simple devices having almost no conflicts, while complex devices in the STM32H7 family have a significant 4 % conflict rate. The opportunity to use majority voting also decreases with complex devices.

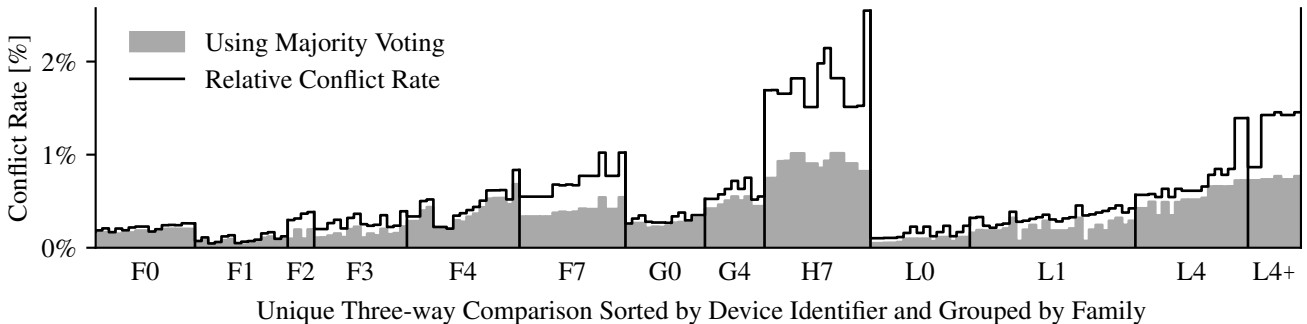

Figure 8: The distribution of bit field conflicts is spread wider than for the registers; however, it compensates with an even higher use of majority voting across all devices.

When we aggregate the register conflicts into the top five associated peripherals as compiled in the appendix (Table 13), the STM32H7 devices alone are responsible for over half of the observed conflicts. Comparing the three sources manually, we discovered a lot of colliding register locations to have aliases with differing bit fields depending on the peripheral runtime configuration. While our ④ reference manual pipeline extracts these location aliases faithfully, the ⑥ CMSIS-SVD files do not always encode these aliases correctly, and the ⑤ CMSIS headers compromise by using a single neutral name instead of a C union. Our linear memory map model does not model multiple names per location, instead keeping only the last name of register or bit field aliases, which can lead to artificial conflicts.

We conclude our evaluation with a summary of all the compared datasets in Table 8. **Our implementation was able to match existing machine-readable data sources both in quantity and quality with a high accuracy of 96.5 %.** We took care to eliminate systemic problems in our pipelines by validating the consistency of our results and finding justifications for outliers manually.

## 5.5   Discussion of Findings

Overall, we studied four aspects related to technical documentation: accessing them, processing their content, encoding the extracted information, and assessing the resulting quality.

We accessed the technical documentation using a custom, vendor-specific PDF parser, implemented with only three times the effort compared to accessing machine-readable sources directly. Still, we expect our modular design to reduce this overhead when adding additional documentation styles from more vendors in the future. The accuracy of the resulting HTML is sufficient to yield reproducible results, with patches required only to repair already-existing formatting issues in the vendor-provided PDFs.

Both the table processing and a simple regex-based text mining interfaces were simple to implement and performed very practicable, even for complex table structures and text fields, yielding highly-detailed datasets. We were able to derive additional context from table captions and surrounding text to further increase our device coverage. However, we found that regex-based text mining is too limiting in practice, as footnotes and text present important information often using different keywords and phrases, making it difficult to write matching patterns for.

We encoded the extracted hardware description data to-

| Dataset | Sources | Method of Comparison | Result | N |
|---|---|---|---|---|
| Device Identifier | | Datasheet ⊇ CubeMX | **93.2 %** | 3024 |
| Package | ③ Datasheet vs. ⑦ CubeMX | Datasheet = CubeMX | **99.68 %** | 2819 |
| Pinout | | Matching pin name at package position | **99.88 %** | 247756 |
| Pin Function | | Matching index for function name at pin | **96.2 %** | 1107035 |
| Interrupt Vector Table | ③ Reference Manual vs. ⑤ Header | Matching vector name at table position | **98.8 %** | 190109 |
| Peripheral | | | **96.4 %** | 42752 |
| Register | ④ Reference Manual vs. ⑤ Header vs. ⑥ SVD | Matching peripheral, register, or bit field name at byte or bit address after majority voting | **97.7 %** | 891044 |
| Bit Field | | | **95.9 %** | 3425903 |
| All Datasets | All Sources | Weighted average over all data points | **96.5 %** | 5910442 |

Table 8: The summary of all data comparisons we performed for this evaluation. The overall quality of the extracted data is very high when compared to the machine-readable sources.

gether with our custom ontology as a knowledge graph. We solved naming differences between sources using only regex substitution patterns. For sources that supported majority voting, we were able to merge machine-readable sources with the technical documentation to detect and repair many data conflicts automatically. The final knowledge graph is easily discoverable using third-party ontology editors, such as Protégé [36], and can be accessed via the owlready2 Python package [37] for integration with code generation tools.

Our extensive evaluation demonstrated the ability of our pipelines to extract highly accurate and complete data from the technical documentation, even when compared to (vendor-provided) machine-readable sources. On top, we were able to find issues in the machine-readable source, particularly in the pinouts and pin functions, that would have been very difficult to spot manually. We can use these identified patterns to guide manual patching efforts much more effectively, thereby increasing the quality of all available data sources. Even for the very large and complex register descriptions, we were able to reconstruct a good enough device resolution to match other sources and repair conflicts via majority voting. While we can send patches for machine-readable data to some of the STMicros GitHub repositories [58–60], to the best of our knowledge, a process to report issues or even encode patches for the PDF documentation is missing so far, making our HTML patches the only known mechanism to do so reliably.

**These results give us high confidence in using our pipeline for extracting data without machine-readable counterparts and achieving similar accuracy**, at least for STMicro technical documentation. However, the architecture of our pipelines and evaluation code is flexible enough to accommodate new data sources and vendors in the future.

## 5.6 Impact of our Tool

The overall accuracy and flexibility of our processor remove the bottleneck of manual data aggregation for many previously described use cases in Section 3. Projects that were limited by accurate PDF access, such as Datasheet2SVD [42], only need to *port* the PDF → HTML and relevant subsequent pipelines instead of starting from scratch. Once the vendor port is complete, it creates an opportunity to extract *significant* amounts of tabular data with very little effort due to the similar tabular data structures across a vendor's product line. Projects based on machine-readable data, such as the modm-devices [33] and embassy-rs [61] projects, can now collaborate on a common data processor (cf. Figure 1), reducing implementation effort and sharing new fixes and features.

The extraction of data also applies beyond STM32 micro-controllers: For example, about 40 % of the already converted STMicro datasheets describe sensors, storage, and communication devices, which are interfaced through external busses and controlled through registers. The I2CDevLib [26] and Cyanobyte [19] projects would greatly benefit from adapted HTML → X pipelines to extract device properties, create SVD register maps, and state machines to generate drivers. Since HTML is such a widely used format, projects that previously relied on custom PDF parsers for hyper-specific tasks, such as uConfig [41] extracting package pinouts, can implement their data extraction directly on HTML even without using our subsequent pipelines. This flexibility can foster an ecosystem of independent data extraction pipelines that can be integrated back into our processor if needed.

Finally, in addition to this evaluation in March 2023, we performed a previous evaluation in July 2022. While STMicro released a total of 63 new datasheets and 18 new reference manuals, our pipeline only required very minor adaptation, mostly related to updating the HTML patches. The total number of compared data points increased by about 1.7 %, yet the relative quality presented in Table 8 varied by less than 0.3 percentage points. In addition, all intermediary results of the individual comparisons (presented in Section 5.4) were very similar, **strongly hinting that our processor implementation will remain stable long-term with very few maintenance requirements, ensuring the usefulness of our tool.**

# 6 Conclusion & Future Work

Our evaluation has underlined the benefits and potential of our tool—a modular processor for automatically extracting hardware descriptions from PDF technical documentation. In the following, we first conclude this paper before presenting alternative concepts and orthogonal pointers for future work.

## 6.1 Conclusion

Our data processor presents a significant improvement over existing generic information extraction solutions when applied to technical documentation due to a specialized PDF parser, unsupervised and fast operation, domain-specific data encoding, and highly accurate results. Our extensive evaluation results confirm that our pipelines are free of systemic issues and can extract information relevant to existing code generation use cases with very high accuracy. In addition, our tool is immediately applicable to a number of open-source projects that would significantly benefit from its use.

With over a thousand captioned tables in our HTML archive of STMicro technical documentation, they showcase the significant potential for extracting data not available in a machine-readable format and therefore provide access to information only published in the documentation. We expect many *entirely new* use cases to be made possible with our tool that are unrelated to code generation: comparing revisions of PDF documents in normalized HTML form, linking code to sections of documentation automatically, using constraint solvers to find layout solutions for pin signal usage in electronic design, validating compatibility of existing software drivers with newly released microcontrollers.

## 6.2 Future Work

We anticipate three specific improvements to our tool that increase its usefulness even beyond our usage:

**Text Mining.** Our implementation uses regex pattern matching to find information in text, which is a brittle and error-prone manual process. However, since technical documentation uses a lot of domain-specific jargon with many nouns and abbreviations only making sense in context, the use of traditional text mining approaches may be difficult. A custom semantic parser could first be seeded by the data extracted from table processing to make domain-specific text mining possible.

**Processing Figures.** For our work, we only focused on tables and text in the technical documentation since table processing is a well-researched approach, and the table content mirrored the data in the machine-readable source most accurately. However, in addition to the 152 041 tables, the STMicro datasheets and reference manuals also contain 52 568 figures. Other applications could benefit from processing these figures since some data, especially state machine diagrams and board pinouts, is only encoded graphically. We could convert the Postscript-based graphics into scalable vector graphics to preserve the vector and text data [16] so that it can be programmatically accessed later on. For example, quasi-regular tabular structures in figures can be found using whitespace analysis [6, 11, 47] or by interpreting text in geometric shapes such as rectangles as imaginary table cells, which allows using table processing for content extraction. However, more complex figures will require more specialized approaches with a better understanding of graphics rendering.

**Machine Learning.** Mathematical formulas are a special case of graphics mixed with text, e.g., $\sqrt{2}$, which our implementation cannot correctly detect or convert. An approach could be to convert them to a markup language such as MathML [67] using a machine-learning model [13]. More complex, cutting-edge projects using machine learning to access PDFs and technical documentation are currently limited to using the text only [7, 8]. Our tool could help infuse these solutions with relevant semantic information via our knowledge graph.

There are many more opportunities for future work to build on our data processor to extract information from technical documentation in other fields than just for firmware development. Our tool has laid the foundation for developing challenging new use cases that require automatically processing technical documentation and thus provides a vital contribution to improving the porting process of embedded software. We will actively maintain our tool in an open-source project to foster its evolution.

## Acknowledgement

## Artifacts

The artifacts are currently under evaluation by the artifacts evaluation board.

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

## A   Appendix

In addition to our evaluation in Section 5, we inspected several data points manually to confirm that our processor was free of systematic issues that could skew our data sets. Here, as an addendum, we present a selection of these manual evaluations based on the STMicro input sources from March 2023.

### A.1   Pinout

The device pinouts we extracted from the technical documentation in Section 5.4.3 matched the STM32CubeMX data very accurately at 99.88 %, with only 20 discrepancies. We, therefore, manually investigated these issues by comparing the PDFs with the raw STM32CubeMX database, where we found formatting issues and confused packages to be the main source of errors. The results of our manual evaluation are encoded in Table 9 and Table 10.

### A.2   Pin Functions

When we evaluated the pin functions in Section 5.4.4, we noticed that the STM32L1 device family had a significantly higher relative amount of conflicts than the rest, as shown in Figure 11. Since this amount of conflicts was too much to manually inspect, we instead investigated the most prominent patterns that emerged, in particular, the conflict of an analog or special hardware function ("additional function") with a digital signal multiplexer ("alternate function"). We verified that the hardware implementation of the alternate functions is identical across all compared devices; therefore, these conflicts are easy to detect since an analog signal cannot be routed through the digital multiplexer and neither the other way around. When we applied these assumptions to our data, we found that particularly the STM32L1 family suffers from systemic conflicts of only a few analog functions, which were mapped wrong across the whole device range, explaining why the family is such an outlier. A selection of prominent patterns, including their explanations, is presented in Table 12.

### A.3   Register Descriptions

During the comparison of the register descriptions in the reference manuals, CMSIS header, and CMSIS-SVD files in Section 5.4.5, we noticed a significant amount of register conflicts in the STM32H7 devices, as listed in Table 13. Comparing the three sources manually, we found a lot of register locations to have aliases with differing bit fields depending on the peripheral runtime configuration. We also noticed slightly different names in the reference manuals than in the other sources; however, the bit field structures and their functionality appears to be compatible, according to the associated

| Affected Devices | −Missing, +Added, =Renamed Positions | Cause of Issue in Datasheet Table and Figures |
|---|---|---|
| STM32G431CBYx | −A4, +A43 | Typo for position A4, figure is correct. |
| STM32L412TBY6P | −E5 | Typo for position F5, figure is correct. |
| STM32H745XxHx, STM32H747XxHx, STM32H755XxHx, STM32H757XxHx | +VDD | VDD name is placed into the position column instead of name column. |
| STM32H747ZIY6 | −A13 | Missing pin position, figure shows A13=NC. |
| STM32H750XBH6 | −G2, −F1 | Missing pin position, figure shows G2=NC, F1=NC. |
| STM32H757ZIY6 | −A13 | Missing pin position, figure shows A13=NC. |
| STM32L071VxIx, STM32L072VxIx | −E3 | Missing pin position, figure shows E3=VSS. |
| STM32L151QCH6, STM32L152QCH6, STM32L162QCH6 | −K1 | Missing pin position, figure shows K1=OPAMP3_VINM. |
| STM32L053CxUx, STM32L063CxUx | Pins 2...7 renamed | Position cells are shifted down by 1 row. |
| STM32L062C8U6 | −46 | Missing position row, figure shows 46=PB9. |
| STM32L412CBxxP | 22=(PB11, VDD), 45=(PB8, PB9), 46=(PB9, VDD) | Missing both package column and figures for the SMPS package variant. Our pipeline instead uses the closest non-variant match. |
| STM32L562QEI6P | B4=(PG15, VDD12), M11=(PG11, VDD12) | Missing both package column and figures for the SMPS package variant. Our pipeline instead uses the closest non-variant match. |

Appendix Table 9: These pin position and name mismatches are all attributed to mistakes in the datasheet: missing entries, typos in cells, and formatting issues. Our pipeline could not find two packages for devices with a switched mode power supply (SMPS) feature and instead used the closest non-variant match.

textual descriptions. We imagine these issues to be the results of the large complexity of the peripherals, having many registers with similar names and interlocking functionality.

| Affected Devices | −Missing, +Added, =Renamed Positions | Cause of Issue in STM32CubeMX Database |
|---|---|---|
| STM32F038E6Y6 | E2=(PB1, NPOR) | Wrong entry, datasheet table and figure both show E2=NPOR. |
| STM32F048TxY6 | D2=(NPOR, PB1), F2=(PB1, NPOR) | Wrong entry, datasheet table and figure both show D2=PB1 and F2=NPOR. |
| STM32L452REYxP | 29 renamed pins | Uses non-variant instead of SMPS package. |
| STM32L476QxIxP, STM32L4P5QxIxS, STM32L4R5QxIxP | C6=(PG14, VDD12), L11=(PB11, VDD12) | Uses non-variant instead of SMPS package. |
| STM32L476QxIxP, STM32L4P5QxIxS, STM32L4R5QxIxP | C6=(PG14, VDD12), L11=(PB11, VDD12) | Uses non-variant instead of SMPS package. |
| STM32L4R5AII6P | C6=(PG15, VDD12), M10=(PH11, VDD12) | Uses non-variant instead of SMPS package. |
| STM32L552QEI6 | B4=(V15SMPS, PG15), M10=(VLXSMPS, PG13), M11=(V15SMPS, PG11), M9=(VDDSMPS, PG14) | Uses SMPS instead of non-variant package. |
| STM32L552VET6 | Pins 20…51, 98, 99 renamed | Uses SMPS instead of non-variant package. |
| STM32L552ZETx | Pins 31…73, 126…143 renamed | Uses SMPS instead of non-variant package. |

Appendix Table 10: The STM32CubeMX database is often using the wrong package for devices with an optional switched mode power supply (SMPS) feature as indicated by the variant key in the identifier. As shown in the first two rows, only three other pins were simply wrong, with the rest of the data matching the datasheet.

| Family | Number of Functions | Number of Conflicts | Absolute Rate of Conflicts | Relative Rate of Conflicts |
|---|---|---|---|---|
| STM32H7 | 215442 | 10184 | 24.2% | 4.7% |
| STM32L1 | 30124 | 6037 | 14.3% | **20.0%** |
| STM32G0 | 66469 | 3960 | 9.4% | 6.0% |
| STM32L4 | 174531 | 3889 | 9.2% | 2.2% |
| STM32F4 | 130518 | 3750 | 8.9% | 2.9% |
| STM32F7 | 114590 | 3130 | 7.4% | 2.7% |
| STM32F0 | 29487 | 2430 | 5.8% | 8.2% |
| STM32G4 | 85415 | 1539 | 3.7% | 1.8% |
| STM32F2 | 22151 | 1411 | 3.4% | 6.3% |
| STM32L0 | 62437 | 1220 | 2.9% | 2.0% |
| STM32F3 | 43468 | 1052 | 2.5% | 2.4% |
| STM32U5 | 52067 | 1047 | 2.5% | 2.0% |
| STM32L5 | 19778 | 858 | 2.0% | 4.3% |
| STM32WB | 8487 | 677 | 1.6% | 8.0% |
| STM32WL | 5888 | 384 | 0.9% | 6.5% |
| STM32H5 | 42602 | 321 | 0.8% | 0.8% |
| STM32C0 | 3581 | 181 | 0.4% | 5.1% |

Appendix Table 11: The conflict rates of pin functions sorted by absolute rate. The STM32H7 devices have the largest amount of pin functions and, therefore, also the largest absolute share of conflicts, while their relative rate of about 5 % is comparable to other families. Meanwhile, over one-fifth of the pin functions of the STM32L1 family conflict, pointing to a systemic data issue.

| Occurances | Functions | Conflict | Description and Cause of Issues |
|---|---|---|---|
| 654
23 | COMPx_INP
COMPx_INM | A≠14 | Comparator input is analog, STM32CubeMX database is wrong for the entire STM32L15x family. |
| 446 | TIMx_ETR | 1≠A | Digital signal where the STM32CubeMX database is wrong for the entire STM32L1 family. |
| 514
319 | SYS_WKUP
SYS_TAMP | A≠0 | Special digital input signal hardwired into PA0 pin to wake up from deep sleep and tamper detection. STM32CubeMX database is wrong for the entire STM32L1 family. |
| 497
497 | RCC_OSC_IN
RCC_OSC_OUT | A≠0 | Special analog signal that must be configured by RCC peripheral. This is a datasheet issue on some STM32F2/F4 devices and a STM32CubeMX database issue on STM32L1 family. |
| 296
105
93
69 | UCPDx_FRSTX | 6≠A
4≠A
0≠A
1≠A | Digital signal that is wrong in the STM32CubeMX database for the entire STM32G0 family. |
| 258
256
128
128
128
108
91 | TIMx_BKIN | 1≠3
13≠3
1≠12
12≠2
12≠3
1≠14
2≠3 | Digital signal collide on alternate function index, as the STM32CubeMX database for the STM32L4 and STM32L5 family contains two functions TIMx_BKIN and TIMx_BKIN2 that seem to have gotten confused. |

Appendix Table 12: A selection of the most interesting and common patterns of pin function conflicts. All of these should have been easy to catch, even without a comparison with other sources, by simply validating how digital vs. analog signals are multiplexed.

| Family | Share of Conflicts | Top 5 Peripherals with Register Conflicts | | | | |
|---|---|---|---|---|---|---|
| STM32F0 | 0.2% | 100% DBGMCU | | | | |
| STM32F1 | 0.5% | 62% FSMC | 12% ADC | 12% USB | 6% SDIO | 3% CEC |
| STM32F2 | 1.7% | 42% ETH | 32% USB | 14% FSMC | 11% ADC | |
| STM32F3 | 1.1% | 41% ADC | 32% HRTIM | 20% EXTI | 4% CEC | 4% I2C |
| STM32F4 | 5.4% | 20% USB | 18% I2C | 15% FSMC | 14% DFSDM | 10% QSPI |
| STM32F7 | 12.6% | 48% DFSDM | 19% USB | 12% FSMC | 8% DSI | 3% ADC |
| STM32G0 | 2.0% | 56% DMA | 11% SYSCFG | 12% EXTI | 8% UCPD | 6% COMP |
| STM32G4 | 3.8% | 40% DMA | 24% HRTIM | 10% ADC | 9% FSMC | 7% UCPD |
| STM32H7 | 60.2% | 23% DMA | 17% DFSDM | 16% RAMECC | 10% ETH | 9% HRTIM |
| STM32L0 | 0.8% | 61% FLASH | 26% COMP | 13% SYSCFG | | |
| STM32L1 | 1.2% | 59% RI | 26% FSMC | 8% RTC | 7% OPAMP | |
| STM32L4 | 3.9% | 63% DFSDM | 18% RTC | 10% FSMC | 6% USB | 4% DAC |
| STM32L4+ | 6.6% | 36% DFSDM | 30% DSI | 11% FSMC | 8% DMA | 5% USB |
| Total | 100% | 22% DFSDM | 17% DMA | 11% USB | 9% RAMECC | 7% HRTIM |

Appendix Table 13: The STM32H7 family is responsible for the majority of register conflicts. The peripherals with the most conflicts are all very complex, which probably contributes to the issue in general.

