# OpenReview forum: "Tool: Automatically Extracting Hardware Descriptions from PDF Technical Documentation"
_JSYS/2023/May_Papers — JSYS 2023 May Papers Submission_

### Official Review · Reviewer_d3sE · 2023-05-28

**Decision:**

Weak accept: good paper with flaws that can be fixed in three months

**Strengths:**

* The implementation sounds complete.
* The accuracy achieved by the tool is impressive.
* Finding discrepancy between the data sheet and ST's SVD descriptions is very impressive too! * Have you reached out to ST Microelectronics regarding these discrepancies?

**Weaknesses:**

* It would have been nice to have data for more than 1 vendor to truly evaluate the strength of the implementation.
* In the age of ChatGPT, it is really important to understand how much of the parsing can be offloaded to a Transformer instead of relying on Table parsers and the like. I would like to see a section from the authors on the suitability (or lack thereof) of ML techniques in the parsing aspect.
* I encourage the authors to create:
   (1) a new semantic format for Datasheets based on their work
   (2) a tool which can automatically convert PDFs to this new format which vendors can use to do it and internally validate it prior to releasing it.
A tool of this kind is really in need.

**Detailed Comments:**

The paper has a few typos and possibly needs a re-read by someone who has not read it before:
* Page 1: "Hardware vendors typically provide a HdS implementation in the C programming language only" Need full stop after "only"
* Page 1: "newer compiled languages and dynamic runtimes, such as [32, 35, 52, 72],": Please list the name of the language and then put the reference number next to the name. Need to go to the references to know which languages you are referring to.
* Reference URL for [38] is no longer valid.
* Page 8: "We removed all devices for which no datasheet or reference manual exits,": I suppose you meant to say "exists".
* Figure 5 / 6: The description is not very clear and is difficult to read. It took me a few passed to understand what you were trying to say.

**Expertise:**

Follow the literature closely, last published 5+ years ago

**Summary Of Review:**

* The paper outlines the design and evaluation of an automated way to parse data sheets of various micro-controllers and make them available in a format which makes it easy to generate HAL code per microcontroller. The use of PDFs for publishing datasheets makes them unsuitable for automatic code generation. From the reading of the paper it seems that a complete implementation is available for 1 vendor.
* The paper is written more from a software design perspective for a practical problem and the paper does not highlight a specific research problem that has been addressed. It is more product focussed. I would urge the authors to further extend their work and propose a more structured format for data sheets (instead of the current PDF format). Of course, this would require them to create a structured format covering all aspects of the data sheet and not limited to register descriptions, interrupt vector table etc.

**Usefulness:**

No

---

### Official Review · Reviewer_RpNT · 2023-06-05

**Decision:**

Weak accept: good paper with flaws that can be fixed in three months

**Strengths:**

1. Flexible and easily adaptable design
The authors recognize the current gaps in the HdS stack and propose a solution that can either be used in its entirety or portions of the data processing pipelines can be selectively utilized for applicable use cases. Though they set out to mainly target to close gaps in code generation in an HdS stack, it presents opportunities for others to explore and build on this work in other areas.

2. Extensive background research and thorough literature survey
The authors have provided a comprehensive summary of previous research in the area, enumerated the existing tools for each phase of their pipeline, their workings, applications and shortcomings. Thus, establishing a need for a modular end-to-end data processor to provide domain-specific data and enable code generation of embedded system software with less manual labor.

None of the individual parts of their proposed design is novel. Since they establish their intention to solve an HdS challenge at large and not attempt to build the best or fastest tools for each phase in their design, they make a good case for leveraging already available tools.
In that regard, they have effectively identified relevant tools that directly apply and build on these findings to design an automated tool.

3. Detailed analysis of the design
The authors have done good justice in presenting the evaluation of their data processing pipelines. They address several challenges they faced in their qualitative analysis and also explain the approach they took to solve - such as naming conflicts, inconsistent data fields that they resolved by employing majority voting technique and further provide an in-depth memory map size comparisons to demonstrate the effectiveness of the information from reference manuals.

4. Extensibility and discussion of drawbacks of proposed design
While the authors provide convincing data to prove the usefulness of their modular processor as a first-step in the design space, they also identify the limitations of their tools and freely discuss the opportunities for improvements. They scrutinize the fragile applicability of their regex matching approach, and provide ideas to extend text mining to become aware of domain-specific nomenclature. They also speak to the absent integration of processed graphics from technical documentations, providing a strong direction for future work that others can contribute towards.

**Weaknesses:**

1. The authors only briefly talk about OWL and the knowledge graphs. How was the OWL generated from HTML, CMSIS SVD and STM32CubeMX DB? It would help others to follow their methodology if authors elucidate the OWL conversions in more detail. In general, lot of emphasis has been laid out on their analysis and less on implementation. Strongly encourage the authors to consider showing the steps through examples, include code-snippets to enable readers better grasp the idea
2. They provide the processing times of each step of the pipeline - where the PDF->HTML conversion severely outdoes the rest. Did they evaluate alternative tools to examine if there is an opportunity to do better? Also, provide reasoning behind the choice of specific tools in their design
3. They aim to develop a fully automated tool to extract hardware information. However, the authors mention they had to manually fine-tune and apply patches to adjust for formatting issues during the PDF->HTML conversion. Did they explore options to detect and fix the issues in a more structured way? This seems to entail a lot of manual labor to sift through. Since the focal point of the design is to automate the entire flow, they need to attempt to solve this/provide ideas.

**Detailed Comments:**

1. Section A.2: First sentence references Figure 11. Should be updated to Table 11
2. Section 1: Contributions: Based on the outcomes of this evaluation, internal consistency to establish a method to merge multiple sources and arbitrate conflicts based on qualitative metrics -> better articulate this sentence. seems a bit incoherent
3. Section 1: small grammatical error
Hardware vendors typically provide a HdS implementation in the C programming language only. Add a period at the end of sentence




**Expertise:**

Follow the literature closely, last published 5+ years ago

**Summary Of Review:**

The authors propose a design for a modular end-to-end processor to extract datasets from technical documentation released by vendors in PDF format and leverage it for code generation tasks, along with information machine-readable sources. Its an interesting paper, with detailed background research write-up on the state-of-the art tools available for each phase of their design. They explain the need for their design and the shortcomings of existing frameworks and reveal new meaning to their work.
The paper also clearly delineates the methodology used for evaluation and delves into the comparative analysis of the design against machine-readable results with enough depth. The challenges in register name conflicts, inconsistency in data between multiple sources and solving it through majority voting approach are well-addressed.
Further, showing the memory map sizes for the various sources and the resulting comparative analysis visualization are effective instruments to show the data distributions to the readers. Overall, its a well-written paper that puts forth the design efficiently.
However, for a work that sets out to automatically extract data, there seems to be notable manual efforts involved in identifying and explaining inconsistencies, fixing formatting issues during conversions, etc. Also, while its commendable efforts on the authors end to provide a detailed analysis of their work, I'd encourage them to also explain a few stages involving OWL in their implementation in more detail.

**Usefulness:**

Yes

---

### Official Review · Reviewer_67eN · 2023-06-07

**Decision:**

Weak accept: good paper with flaws that can be fixed in three months

**Strengths:**

- The introduced processor can potentially be useful to produce machine-readable resources from technical documentation, needed by code generation platforms.

- The modular design of the processor offer flexibility with regard to maintenance and future improvements.

- The paper is well-written and follows a logical structure.

**Weaknesses:**

-  The performance of the introduced framework has been only assessed by comparing the extracted information with already existing machine-readable resources. By comparing the accuracy and performance with existing tools, the confidence in the high quality and efficiency of the introduced processor can be enhanced. As a suggestion on enhancing the evaluation confidence is it possible to compare the performance of the suggested processor with uConfig at least on a limited basis on the device pinout extraction task?

Such a comparison would provide valuable insights into the superior performance and reliability of the processor, further validating its capabilities and potential benefits.

- Some details with regard to the algorithms used in each major pipelines are missing. As an example it is unknown how the potential conflicts are being resolved during the evolving of OWL. (page 6. OWL--> OWL section).

- On page 7 (PDF->HTML quality section), it has mentioned that manual comparison between inputs and outputs was needed to discover formatting issues. As the processor is intended to be an automatic tool for information extraction and the evaluation has only made based on STMicro documents, how it can be guaranteed that no more manual adjustments are needed when applied to other vendors technical documentations ?

**Detailed Comments:**





**Expertise:**

Please contact the Area Chair if your expertise is lower than this

**Summary Of Review:**

This paper introduces a modular PDF processor to extract detailed hardware descriptions from technical documentations (Datasheets and User Manuals). It is specifically focused on converting tabulated information into knowledge graphs to be used by hardware-dependent software (HdS) code generation platforms. The design and internal pipelines of the processor is explained and the interconnections have been described. Finally the accuracy of the processor is evaluated based on the existing machine-readable STMicro resources provided for STM32.

**Usefulness:**

Yes

---

### Official Review · Reviewer_w4Tr · 2023-06-15

**Decision:**

Weak accept: good paper with flaws that can be fixed in three months

**Strengths:**


1) The paper is written very well and reads easily.  Although concise, lots of effort has been put to give detailed explanation of each section which helps to easily understand scope and contributions of the work

2) The tool's functionality is verified by comparing the data extracted from the real technical documentation against existing machine-readable sources. The authors also provide a detailed analysis of the quality, trustworthiness, and completeness of each data source.

3) I think the biggest advantage of this work is that the extracted data is encoded as a knowledge graph via a custom ontology. In this way it will be widely accessible for various use cases and it can trigger new research directions in this field.





**Weaknesses:**

1) The effectiveness of the tool is dependent on the quality and consistency of the source documents. It would be interesting to see results if the technical documentation is incomplete. How does this could impact the quality of the extracted data? In general, how this tool works with documents that do not provide enough information or they have missing informations?

2) While the paper has two sections on discussing the significance of the research i.e,, "Discussion of Findings" and "Impact of our Tool", it is not clear how it can help with future research in the same area. I think authors should have one more section in paper and discuss about potential future research ideas.

**Expertise:**

Published in this area in the last 5 years

**Summary Of Review:**


This work addresses the challenge of porting hardware-dependent software (HdS) stacks in embedded microcontrollers and tries to ease the burdensome task of extracting device-specific data from technical documentation, often only available in PDF format, and converting it into code, a labor-intensive process which is typically manual which can slows down new HdS projects.

The authors propose an automated, unsupervised process to extract data from multiple sources, including technical documentation, and convert it into a structured, machine-readable format. This data can then be used for code generation tasks for multitude of programming languages, significantly reducing development efforts.

Overall I find the work very interesting and beneficial for the community. I hope the authors will open source the developed tool (as mentioned in the paper) as I’m sure the embedded systems community will strongly benefits from this work.

**Usefulness:**

Yes

---

### Meta-Review · Area_Chair_bWcr · 2023-06-17

**Recommendation:** Revise
**Confidence:** 5

**Metareview:**

This paper presents a modular processor designed to extract detailed data sets from technical documentation in PDF format, specifically targeting microcontroller hardware descriptions. The extracted data can be used for code generation in the embedded software domain. The paper received multiple reviews with varying opinions on its strengths and weaknesses. The reviews are, in general, positive but point out shortcomings that must be addressed in order for your paper to be accepted. To this end, my recommendation to the EiC is that you carry out a major revision in which you address every concern raised by the reviewers. When submitting it for the second round of review, please provide a bulleted list of how you addressed each concern, with clear reference to the specific section in the revised paper. The final decision will be made based on these changes.

Isolating major concerns:

- A section on the suitability (or lack thereof) of ML techniques in the parsing aspect would add to the quality of the manuscript.
- Propose a more structured format for data sheets (instead of the current PDF format) covering all aspects of the datasheet. Provide a structured approach to automatically convert PDFs to this new format, which vendors can use to do it and internally validate it prior to releasing it.
- One of the major reviewer concerns relates to the tool's effectiveness when dealing with incomplete technical documentation and the need for a section discussing potential future research ideas.
- Enhance the evaluation confidence by comparing the performance of the proposed processor with other tools, particularly uConfig, and provide more details about the algorithms used in each major pipeline. Discuss quantitatively and qualitatively how the tool would handle technical documentation from vendors other than STMicro.

---

### Decision · Program_Chairs · 2023-06-18

**Decision:**

Revise

**Comment:**

Dear authors,

Thank you for submitting your work to JSys. Please accept our apology for the slight delay in the decision.

All reviews are now public. The reviews converged to a "Revise" decision. Please refer to the [Instruction for authors](https://www.jsys.org/instructions#submitting-a-revision) for details about the procedure. Please, do not hesitate to let us know if anything is unclear.

The meta-review outlines the main points that the reviewers would like to see addressed in the revision. Please use the "Official Comment" button to interact with the area chairs/editors/reviewers if you need or wish to do so.

Best,
Romain,
JSys Editor-in-Chief